# Transcriptomic Signatures and Upstream Regulation in Human Skeletal Muscle Adapted to Disuse and Aerobic Exercise

**DOI:** 10.3390/ijms22031208

**Published:** 2021-01-26

**Authors:** Pavel A. Makhnovskii, Roman O. Bokov, Fedor A. Kolpakov, Daniil V. Popov

**Affiliations:** 1Institute of Biomedical Problems of the Russian Academy of Sciences, 123007 Moscow, Russia; maxpauel@gmail.com (P.A.M.); romanbokov94@yandex.ru (R.O.B.); 2Institute of Computational Technologies of the Siberian Branch of the Russian Academy of Sciences, 630090 Novosibirsk, Russia; fkolpakov@gmail.com; 3Faculty of Fundamental Medicine, M.V. Lomonosov Moscow State University, 119991 Moscow, Russia

**Keywords:** transcriptome, disuse, transcription factors, heterodimer, extracellular matrix, mitochondrion, skeletal muscle, exercise

## Abstract

Inactivity is associated with the development of numerous disorders. Regular aerobic exercise is broadly used as a key intervention to prevent and treat these pathological conditions. In our meta-analysis we aimed to identify and compare (i) the transcriptomic signatures related to disuse, regular and acute aerobic exercise in human skeletal muscle and (ii) the biological effects and transcription factors associated with these transcriptomic changes. A standardized workflow with robust cut-off criteria was used to analyze 27 transcriptomic datasets for the vastus lateralis muscle of healthy humans subjected to disuse, regular and acute aerobic exercise. We evaluated the role of transcriptional regulation in the phenotypic changes described in the literature. The responses to chronic interventions (disuse and regular training) partially correspond to the phenotypic effects. Acute exercise induces changes that are mainly related to the regulation of gene expression, including a strong enrichment of several transcription factors (most of which are related to the ATF/CREB/AP-1 superfamily) and a massive increase in the expression levels of genes encoding transcription factors and co-activators. Overall, the adaptation strategies of skeletal muscle to decreased and increased levels of physical activity differ in direction and demonstrate qualitative differences that are closely associated with the activation of different sets of transcription factors.

## 1. Introduction

Skeletal muscle tissue makes up more than a third of the adult human body mass playing a key role in fat and carbohydrate metabolism and positively regulating the functions of various tissues by acting as a secretory organ [1]. Inactivity and chronically reduced physical activity are widespread in most Western countries and are strongly associated with reduced life span and the development of numerous disorders, such as type 2 diabetes mellitus and other metabolic disorders, cardiovascular diseases, depression and other neurological diseases, chronic fatigue syndrome, cachexia and sarcopenia [1,2,3,4]. Regular physical activity, especially aerobic exercise, is broadly used as a key intervention to prevent and treat these pathological conditions (reviewed in [5]). Therefore, understanding the molecular mechanisms underlying skeletal muscle (dis)adaptation to inactivity and aerobic exercise is of fundamental importance.

Examination of transcriptomic responses to an intervention provides a unique opportunity to assess the directions of adaptive changes and predict the underlying molecular mechanisms. Specifically, transcriptomic approaches can be used to (i) evaluate changes in the expression levels of genes of interest, (ii) predict biological effects associated with the expression of multiple genes with known and unknown functions and (iii) predict upstream regulators associated with the transcriptomic responses.

Several transcriptomic studies have examined the effects of disuse and aerobic exercise on human skeletal muscle; however, the results of these studies sometimes demonstrate low concordance, even in terms of differentially expressed genes (Table 1). This discordance may be related to the use of small and homogeneous groups that do not represent the general population, as well as differences in the feeding status, experimental model used, experimental design, diurnal time of muscle biopsy sampling, quality of sample preparation, statistical approach and cut-off criteria (P-value adjusted and fold-change). Using a standard workflow to reanalyze the raw data from multiple studies investigating the transcriptomic responses to disuse and aerobic exercise provides an opportunity to improve the depth and significance of the findings and to increase the total sample size and statistical power.

There have been several attempts to systemize human skeletal muscle transcriptomic data [6,7,8,9]. In the most comprehensive recent study [9], the transcriptomic changes related to disuse, different types of long-term training (aerobic, high intensity interval and strength), and acute exercises were simultaneously analyzed in various human skeletal muscles. The study searched for differential expression of genes common for various interventions. Four genes (*DNAJA4*, *KLHL40*, *NR4A3* and *VGLL2*) have been shown to play a role in the adaptation to disuse, acute aerobic exercise and resistance exercise; *NR4A3* was identified as one of the most exercise- and disuse-responsive genes. In contrast to the previous study, we attempted to (i) identify and compare the transcriptomic signatures related to disuse, long-term/regular aerobic exercise training and acute aerobic exercise in a muscle (the vastus lateralis (VL) muscle) and (ii) predict and compare the biological effects and upstream regulators (transcription factors (TFs)) associated with these transcriptomic changes. For this purpose, we used a standardized workflow to analyze 27 transcriptomic datasets and to identify differentially expressed genes (DEGs) for the VL muscle of healthy humans subjected to disuse, long-term aerobic training and acute aerobic exercise (Table 1). Then, for each intervention, DEGs related with each dataset were integrated using a rank aggregation method [10] with a robust cut-off criterion (P_adj_ < 0.01). In the results, we identified several hundred genes related to each intervention and, for the first time, examined the dynamics of the transcriptomic responses to acute aerobic exercise (Table 1). This approach allowed us to predict the TFs associated with the different interventions (via a TF binding site enrichment analysis) and describe the strategy of (dis)adaptation of human skeletal muscle to decreased and increased levels of physical activity.

**Table 1 ijms-22-01208-t001:** Characteristics of raw data sets included in the meta-analysis and number of up- (UP) and down-regulated (DOWN) protein coding genes in each individual study and in the meta-analysis. Datasets showing no up- and down-regulated genes were not included in the meta-analysis.

GEO ID	Model and Reference	Duration (d or wk)/Recovery Time (h)	N of Subjects (Males/Females)	Mean Age (Range)	Mean BMI	Muscle	Method	Individual Study: N of UP/DOWN Genes	Meta-Analysis: N of UP/DOWN Genes
GSE113165	Bed rest [11]	5 d	28 (13/15)	53 (30–76)	23.9	VL	RNA seq	339/696	76/205
GSE126865 GSE130722	Bed rest [12]	10 d	3 (3/0) 5 (5/0)	70 (64–76) 66 (61–73)	29.0 27.6	VL	RNA seq	101/310	
GSE24215	Bed rest [13]	10 d	10 (10/0)	25 (24–26)	24.0	VL	DNA array	47/184	
GSE14901	A leg immobilization [14]	14 d	24 (12/12)	21 (19–23)	23.6	VL	DNA array	479/571	
GSE104999	Bed rest [15]	21 d	12 (12/0)	27 (21–33)	24.0	VL	DNA array	59/170	
GSE33886	A leg immobilization [16]	21 d	6 (-)	21 (21–21)	22.0	VL	DNA array	0/0	-
GSE147494	Cycling training [17]	5 wk	40 (40/0)	22 (20–24)	22.4	VL	DNA array	134/29	374/119
GSE35661	Cycling training [18]	6 wk	24 (24/0)	23 (22–24)	23.0	VL	DNA array	563/142	
GSE27536	Cycling training [19]	8 wk	12 (10/2)	65 (-)	-	VL	DNA array	1212/386	
GSE120862	Cycling training [20]	8 wk 1 leg 8 wk 1 leg	7 (7/0)	21 (21–24)	23.0	VL	RNA seq	2145/1508 2127/1119	
GSE1786	Cycling training [21]	12 wk	6 (6/0)	68 (65–71)	27.0	VL	DNA array	194/88	
GSE60591 GSE60833 GSE60833	Knee extension training [22]	12 wk 1 leg 12 wk 1 leg 12 wk 1 leg	22 (11/11) 11 (6/5) 10 (5/5)	27 (26–28) 28 (25–32)	24.0 25.9	VL	RNA seq	931/322 111/155 58/97	
GSE117070	Cycling training [23]	20 wk	41 (-)	-	-	VL	DNA array	30/61	
GSE111551	Running training	18 wk	13 (13/0)	27 (22–30)	-	VL	DNA array	0/0	-
GSE41769 GSE71972 GSE59363	Acute knee extension [24] Acute knee extension [25] Acute cycling [26]	0 h 0 h 0 h	9 (9/0) 8 (-) 7 (7/0)	52 (47–57) 23 (19–27) 56 (54–58)	26.0 23.7 27.4	VL VL VL	DNA array RNA seq DNA array	218/34 399/703	140/81
								0/0	-
GSE43219 GSE33603 GSE107934	Acute cycling [27] Acute cycling [28] Acute cycling [29]	0.5 h 0.5 h 1 h	14 (9/5) 11 (11/0) 6 (6/0)	33 (23–48) 22 (21–23) 27 (24–30)	26.5 24.5 24.7	VL VL VL	DNA array DNA array RNA seq	456/291 378/134 47/0	272/50
GSE120862	Acute knee extension [20]	1 h	7 (7/0)	21 (21–24)	23.0	VL	RNA seq	802/534	
GSE164081	Acute cycling	1 h	10 (10/0)	32 (30–36)	23.1	VL	CAGE	541/166	
GSE27285	Acute cycling [30]	3 h	8 (8/0)	33 (27–39)	24.0	VL	DNA array	462/320	374/80
GSE33603	Acute cycling [28]	3 h	11 (11/0)	22 (21–23)	24.5	VL	DNA array	932/592	
GSE71972	Acute knee extension [25]	3 h	8 (-)	23 (19–27)	23.7	VL	RNA seq	1411/635	
GSE164081	Acute cycling	3 h	10 (10/0)	32 (30–36)	23.1	VL	CAGE	769/325	
GSE43856	Acute cycling and running [31]	3 h	8 (8/0)	25 (21–29)	23.1	VL	DNA array	0/0	-
GSE59363	Acute cycling [26]	3 h	7 (7/0)	56 (54–58)	27.4	VL	DNA array	0/0	-
GSE107934	Acute cycling [29]	4 h	6 (6/0)	27 (24–30)	24.7	VL	RNA seq	193/20	153/74
GSE86931	Acute cycling [32]	4 h	2 (2/0)	24 (22–25)	22.0	VL	RNA seq	101/45	
GSE87748	Acute cycling [33]	4 h	10 (10/0)	23 (21–24)	23.1	VL	RNA seq	183/173	
GSE120862	Acute knee extension [20]	4 h	7 (7/0)	21 (21–24)	23.0	VL	RNA seq	1295/909	

## 2. Results

Raw data processing revealed that approximately 12,000 to 45,000 genes were expressed in the human VL muscle during each intervention (disuse, regular aerobic training and acute aerobic exercise) (Appendix A). Using the standardized workflow and cut-off criteria (see Materials and Methods) we found that each intervention typically induced expression of several tens or hundreds of genes. However, few or no DEGs were found in a small number of datasets (Table 1) that probably related to technical problems and/or biological reasons (specificity of studies, subjects, etc.). We did not include datasets showing no DEGs in the following analysis (Table 1).

After applying the robust rank aggregation method with cut-off P_adj_ < 0.01, we found that 281, 493 and 815 protein-coding genes were differentially expressed following disuse, regular aerobic training and acute exercise, respectively. Notably, the responses to the different interventions were multidirectional: 73% of the disuse-related genes were down-regulated, whereas 67%–84% of the contractile activity-related genes were up-regulated (Figure 1a, Appendix A). Moreover, each intervention predominantly affected the expression of a specific set of genes (Figure 1b). Surprisingly, pronounced transcriptomic changes occurred immediately after acute exercise (Figure 1a); the changes were observed throughout the 4 h recovery period (Figure 1c).

### 2.1. Regulation of Biological Processes

Next, we used a functional enrichment analysis to identify biological processes that are regulated at the level of transcription following disuse, regular training and acute exercise. This analysis revealed that each intervention was associated with a specific set of biological processes. Specifically, disuse-related down-regulated genes were associated with mitochondrial ATP synthesis and muscle filaments, whereas regular training and acute exercise both up-regulated genes involved in angiogenesis and response to stimulus (Figure 2, Appendix A). In addition, regular training was associated with the up-regulation of genes involved in extracellular matrix organization and cell migration, and acute exercise was associated with an up-regulation of genes encoding transcriptional regulators and those involved in immune response (Figure 2, Appendix A). Notably, we found that different Gene Ontology (GO) terms were enriched at specific times during the acute exercise recovery period: regulation of transcription was a time-specific term, whereas angiogenesis was detected throughout the recovery period (Figure 2).

### 2.2. Expression of Genes Encoding Specific Sets of Proteins

Next, we examined the regulation of genes encoding all proteins related to the metabolic pathways and protein classes identified as being regulated by disuse or exercise in the functional enrichment analysis. Disuse down-regulated the expression of a number of genes encoding enzymes involved in glycolysis/gluconeogenesis (2 of 35), lipid transport and beta oxidation (5 of 26), the tricarboxylic acid cycle (3 of 27) and respiratory complexes (29 of 123), and had a small inhibitory effect on the expression of genes encoding mitochondrial proteins (88 of 1097) (Figure 3a, Table 2 and Appendix A). Interestingly, regular training and acute exercise did not up-regulate the genes encoding these protein groups to any large extent (Figure 3a, Table 2 and Appendix A). In addition, disuse also down-regulated 13 of the 58 genes encoding sarcomeric proteins; notably, 12 of these proteins are expressed predominantly in type I slow muscle fiber.

Regular exercise up-regulated more than half of the genes encoding collagens (15 of 27), as well as a number of those encoding extracellular matrix (ECM) glycoproteins (26 of 122), proteoglycans (6 of 19), ECM regulators (19 of 115), cytokines, growth and secreted factors (20 of 179) and transcriptional regulators (21 of 1527). By contrast, acute exercise mainly induced the expression of genes encoding transcriptional regulators (114 of 1527) and cytokine, growth and secreted factors (20 of 179) (Figure 3b,c, Table 2 and Appendix A).

### 2.3. Transcription Factors Induced by Disuse and Exercise

A TF binding site enrichment analysis (cut-off criteria: FE_adj_ > 1.25 and FDR < 0.05) revealed that several dozen TFs were significantly associated with the genes that were up- or down-regulated following disuse or exercise (Appendix A). Figure 4b,d shows the TFs with binding sites that were highly enriched (FE_adj_ > 1.5 and FDR < 0.05) in the sets of differentially expressed genes. Notably, the opposing directional changes in the transcriptome induced by disuse and exercise were associated with very different sets of TFs. In view of the fact that certain TFs may form various dimers or multimers, we analyzed potential protein–protein interactions to identify clusters of the most enriched TFs. One small cluster of TFs was identified for disuse (ESRRB-SOX17) and for training (FOXA1-PGR-AR) (Figure 4b). In contrast, for acute exercise two small clusters of TFs (SMAD4-MAX and ZSCAN21-ZSCAN22) and two large clusters comprising bZIP TFs belonging to the ATF/CREB/Activator Protein 1 (AP-1) superfamily were identified (Figure 4d).

The relative number of overlapped motifs associated with TFs from the ATF/CREB/AP-1 superfamily in the promoter regions of DEGs was substantially greater than those without confirmed protein–protein interaction (Figure 4e), suggesting that TFs from the ATF/CREB/AP-1 superfamily may act as heterodimers.

## 3. Discussion

Physical disuse and aerobic exercise training markedly affect the phenotype and function of skeletal muscle. In our current meta-analysis, we identified the transcriptomic responses and main biological processes associated with these interventions. Our analysis enabled us to investigate the expression of genes encoding proteins involved in the protein classes (sarcomere, ECM proteins and regulators, cytokine, growth and secreted factors, transcriptional regulators and mitochondrion) and metabolic pathways (carbohydrate and fat metabolism) identified as being regulated by disuse, regular training or acute exercise. Consequently, we were able to elucidate the role of transcriptional regulation in the phenotypic changes related to disuse and exercise.

### 3.1. The Role of Transcription in Phenotypic Changes Induced by Disuse and Training

*Sarcomeric-related proteins.* Bed rest and limb immobilization studies revealed that several days/weeks of disuse progressively decreased muscle volume (~0.4% per day) and strength (reviewed in [41,52]). This decline in the content of contractile proteins is related to a suppression of the protein synthesis rate rather than protein breakdown, at least in the human VL muscle with mixed fiber types (reviewed in [53,54]). During disuse, the human VL muscle shows a decrease in fiber cross sectional area; in contrast to the lower limb muscles, this change is comparable for both fiber types [37,38,39,40,41] and is most likely associated with a (sub-)proportional decrease in the levels of multiple sarcomeric proteins. Using data from a proteomic study [55], we identified sarcomeric-related proteins that are expressed at different levels in fiber types I and II (Appendix A). Our current study found that, in contrast to training, disuse was associated with a reduction in the levels of the mRNAs encoding 13 (22%) of the 58 sarcomeric proteins, 12 of which are expressed predominantly in type I slow muscle fibers (Figure 3a, Table 2 and Appendix A). This finding indicates that transcriptional regulation plays only a partial role in reducing the size of slow type I fibers during the first days/weeks of disuse. Other mechanisms regulating the synthesis of total and specific proteins, as well as protein breakdown, seem to be responsible for reducing the content of other sarcomeric proteins, especially in type II fibers.

*ECM-related proteins.* In the human VL muscle, several days/weeks of disuse impair the mechanical (elongation, stiffness and strain) and material (Young’s modulus) properties of the muscle–tendon unit, as well as the collagen synthesis rate (reviewed in [41]). By contrast, the collagen concentration does not change in response to disuse due to a decrease in muscle mass [43,44,45]. In line with the last finding, we found that only a few genes encoding collagens, ECM glycoproteins and proteoglycans (4 of 168; Figure 3b, Table 2 and Appendix A) were down-regulated following a period of disuse. Among them was *LGI1* (encoding a protein playing a role in suppressing the production of MMP1/3—key enzymes regulating degradation of collagen I, II, III, IV, IX, X, fibronectin, laminin and some proteoglycans), which may be partially related to disuse-induced disturbance of the ECM network.

In the human VL muscle, acute and regular aerobic exercise increases the collagen synthesis rate [56] and concentrations of collagens (reviewed in [46]) and other ECM proteins [35]. Regular aerobic exercise probably has a weak effect on the viscoelastic properties of the muscle–tendon unit [57,58], but substantially reduces exercise-induced muscle damage and soreness (reviewed in [59]) associated with changes in the ECM and endomysium. Here, we found that regular aerobic exercise substantially up-regulated more than a half of genes encoding muscle collagens, as well as a number of ECM glycoproteins and proteoglycans (47 genes in total: 56%, 25% and 32% of the corresponding genes, respectively), including the two most abundant muscle ECM proteins: collagen alpha-1 (I and III) accounting for more than half of the ECM proteins in skeletal muscle [60] (Figure 3b, Table 2 and Appendix A). These responses were predominantly related to changes in the baseline gene expression induced by long-term training and support the notion that ECM proteins are closely regulated at a transcriptional level [35]. Only a few enzymatic ECM regulators are known to be involved in disuse- and training-induced ECM remodeling in skeletal muscle (reviewed in [46,47]). Here, we found that 26 (23%) of the genes encoding ECM enzymatic regulators (metallopeptidases, inhibitors of metalloproteinases, lysyl oxidases, etc.) were up-regulated at baseline after long-term training and acute exercise (Figure 3c, Table 2 and Appendix A). The roles of the majority of these ECM regulators, as well as those of ECM glycoproteins and proteoglycans, in controlling the biogenesis of the ECM in skeletal muscle are still unknown and require additional investigation.

ECM-related proteins play an important role in angiogenesis (reviewed in [61]). Many, but not all [62,63], studies have shown no decreases in the capillary to fiber ratio [64,65,66,67] or capillary density [63,64,65,67] of the human VL muscle after several weeks of disuse. It is difficult to select a complete set of genes related to angiogenesis; however, in line with the studies mentioned above, we found no enrichment of down-regulated angiogenesis-related genes following disuse. By contrast, acute and regular exercise were both strongly associated with an up-regulation of angiogenesis-related genes (Figure 2, Appendix A), which is consistent with the results of previous studies showing an increase in muscle capillary density after a period of aerobic training (reviewed in [42,68]). Collectively, these data demonstrate that angiogenesis is regulated by exercise at the transcriptional level, at least partially.

*Regulatory proteins.* Our meta-analysis revealed that, contrary to disuse, acute exercise massively up-regulated genes associated with responses to various endo- and exogenous stimuli and, especially, with regulation of transcription (Figure 2, Appendix A). To our knowledge, no high-throughput analyses of the effect of disuse on the expression levels of various transcriptional regulator proteins have been performed. Here, we found that disuse had almost no effect on genes encoding TFs, co-activators and repressors (Figure 3c, Table 2 and Appendix A) and speculate that disuse has little effect on the level of these proteins. Hence, disuse-induced changes (mainly decreases) in the levels of muscle proteins appear to occur independently of transcriptional regulators. On the other hand, we found that acute exercise had a powerful effect on genes encoding transcriptional regulators (114 genes), cytokine, growth and secreted factors (20 genes) (Figure 3c, Table 2 and Appendix A), all of which encode proteins that may potentially regulate numerous target genes. Notably, the expression of these regulatory genes was detected immediately after termination of an exercise and peaked to 3 h of recovery (Figure 3c). This change likely serves to increase the levels of the corresponding proteins rapidly, thereby substantially improving the ability of muscle cells to regulate the expression of multiple target genes and to secrete myokines during the latter stages of recovery and/or in response to the next exercise. In line with our data, mRNA encoding TFs have been shown to dominate the earliest responses to various stimuli in human and mouse cells [69]. Collectively, these findings show that the transcriptomic response associated with up-regulation of transcription factor genes to a stress stimulus is conservative and common for cells and tissue.

*Carbohydrate and lipid metabolism.* Disuse negatively regulates the ability of muscle to oxidize fat and carbohydrate, as well as mitochondrial density and function, and thereby decreases aerobic performance (reviewed in [41,49]). By contrast, regular aerobic exercise markedly increases these parameters [42], as well as the concentrations of numerous enzymes related to lipid transport and beta oxidation, the citrate cycle and oxidative phosphorylation, but has no marked effect on the levels of glycolytic enzymes [35,36,50]. Here, we found that disuse and exercise weakly affected glycolysis at the transcriptional level in a multidirectional manner by regulating a gene called *LDHB*. Disuse down-regulated five (of 26) genes related to the transport and degradation of fatty acids. All of these genes are highly expressed and include almost all rate limiting enzymes related to lipid transport (*CPT1B*) and beta oxidation (*ACSL1*, *ACAT1*, *ACAA2* and *HADH*) (Appendix A), indicating that fatty acid metabolism is closely regulated at a transcriptional level during disuse.

Changes in mitochondrial density are related to (sub-)proportional changes in the numbers of mitochondrial proteins. Surprisingly, we found that disuse down-regulated a relatively small number of genes encoding mitochondrial proteins (8% (88 genes) of the complete sets of corresponding genes). In particular, only 11% and 24% of genes encoding the citrate cycle and oxidative phosphorylation enzymes, respectively, were down-regulated in response to disuse. Notably, a previous study found that the same number of genes encoding mitochondrial proteins were expressed at lower levels in sarcopenic human VL muscle showing low mitochondrial density than in age-matched controls [70]. In agreement with our previous findings [20,35], our current analysis revealed that acute and regular exercise induced the expression of only a few genes (2% and 3%) encoding mitochondrial proteins (Figure 3a, Table 2 and Appendix A). It is possible that acute exercise-induced up-regulation of these genes may occur in the latter stages of recovery. However, no massive changes in the expression levels of genes encoding mitochondrial proteins were found 8 h after an aerobic exercise [32]. Meanwhile, the mitochondrial protein synthesis rate has been shown to increase during the first 4 h of recovery [71,72]. Overall, it seems that transcriptional regulation plays only a small role in down-regulating mitochondrial proteins during several weeks of disuse, and seems to have no effect on their exercise-induced up-regulation. The latter may be explained, at least in part, by data showing that mitochondrial biogenesis is associated with increased mitochondria-specific protein synthesis (i.e., translation) [71,72,73], which may be regulated via TISU and TOP motifs in the 5′UTR [74,75] and in a cap-independent manner as discussed previously [35], as well as with chaperone-dependent regulation of mitochondrial proteostasis [35,76,77].

### 3.2. Transcription Factors Induced by Disuse and Exercise

Analysis of the TFs with binding sites that were most enriched (FE_adj_ > 1.5) in the sets of differentially expressed genes revealed that the opposing transcriptional responses to disuse and exercise were associated with very different sets of TFs (Figure 4a–d). Disuse- and training-induced decreases in gene expression were associated with several TFs (mainly ESRRB, HOXC13 and MECP2 for disuse, and PGR, MECOM and AR for training; Figure 4b). A decrease in the expression of specific genes may be explained by the activation of one or more specific transcription repressors. Indeed, some of the TFs associated with the genes that were down-regulated in response to disuse or training can act as repressors. Decreases in gene expression induced by chronic interventions may also be related to a reduction in the activity of transcriptional activators and/or the occurrence of specific epigenetic modifications that limit the ability of these activators to bind to their target sites. For instance, in the human VL muscle, more than 200 genes that are down-regulated in response to regular aerobic training display increased DNA methylation in their promoter regions [78]. To our knowledge, no studies investigating genome-wide epigenetic regulation after disuse have been published to date, but a few studies have shown that disuse increases the methylation of DNA in the promoter regions of some down-regulated genes in human [13] and mouse [79] skeletal muscle.

RELB, JUND, ETV4/5 and TWIST1 were the most enriched TFs associated with genes that were up-regulated after regular exercise (Figure 4d). Notably, in terms of the number of up-regulated genes, the transcriptomic responses to regular and acute exercise were comparable (Figure 1a). However, in contrast to the weak enrichment of TFs after regular exercise, the transcriptomic response to acute exercise was associated with a marked enrichment of multiple TFs at specific time points (including SRF, CREB1/3, CREM, ATF1/2/3/4, FOS, JUN, etc.) and during the 4 h recovery period (SP1, PLAG1, FOSL1/2, EGR1, KLF15, etc; Figure 4d). This finding may suggest that the pronounced up-regulation of genes after regular exercise is regulated partially by mechanisms other than increased TF enrichment. For example, regular exercise-induced increases in the expression levels of approximately 100–250 genes in the human VL muscle occur in parallel with DNA hypomethylation in their promoter regions [78,80].

A protein–protein interaction analysis identified clusters of TFs that were enriched in the disuse- and exercise-induced DEGs. The acute exercise group displayed the two biggest clusters, comprising predominantly bZIP TFs (22 of 28 TFs), which belong to the ATF/CREB/AP-1 superfamily and were strongly enriched immediately and one hour after acute exercise (Figure 4d). Members of the ATF/CREB/AP-1 superfamily share a similar core DNA-binding motif. Indeed, unlike the TFs for which protein–protein interactions were not identified, those identified in the acute exercise cluster demonstrated a marked overlap of motifs (Figure 4e), suggesting that they may act as heterodimers. Indeed, bZIP TFs can form various heterodimeric complexes [81,82,83,84] and can interact with several co-activators (e.g., CREB-regulated transcription co-activators 1/2/3, CREB-binding protein, nuclear receptor corepressor 1, histone deacetylase 3, histone acetyltransferase p300 and KAT5) [85,86,87,88] to enable finely tuned regulation of their DNA-binding specificity and possible genomic targets [83,89,90]. Based on this evidence, we conclude that heterodimerization of TFs belonging to the ATF/CREB/AP-1 superfamily plays an important role in regulating the transcriptomic response to acute exercise. Interaction of various TFs with other TFs and/or co-activators leading to the phase transition is proposed as a key mechanism regulating transcription from a promoter [91]. Our data indirectly confirm this hypothesis for acute exercise and indirectly suggest that other mechanisms regulating transcription from a promoter govern gene expression in the baseline state after disuse and regular training.

### 3.3. Limitations and Further Directions

Only limited numbers of studies examining the transcriptomic responses of muscle to disuse and aerobic training are currently available in the literature. When performing a meta-analysis to detect DEGs related with a specific intervention, the lack of a large dataset markedly increases the probability of false negatives and does not allow a thorough investigation of the dynamics of the transcriptomic responses. In our current analysis, we examined the dynamic changes in the transcriptome that occur up to 4 h after acute exercise; however, studies examining transcriptomic responses during longer recovery periods are warranted. Moreover, only a few studies have investigated the molecular response of human skeletal muscle during re-adaptation after a period of disuse; this topic is an important area for future research. Finally, direct comparisons of transcriptomic data with proteomic, secretomic and other omics data are required to understand the highly complex mechanisms of human skeletal muscle adaptation to various interventions.

The differentially expressed genes identified in our current meta-analysis can be considered as key pieces of the puzzle related to the regulation of gene expression in muscle during periods of disuse and aerobic exercise. To put these pieces into a single picture, mathematical modeling seems to be a perspective approach. For this purpose, we have started to develop a complex mathematical model that links metabolic changes with the expression of some exercise-induced genes (namely *NR4A2*, *NR4A3* and *PPARGC1A*) [92].

### 3.4. Conclusions

Our meta-analysis revealed that the expression levels of several hundred genes are related to long-lasting disuse, regular aerobic training and acute exercise, and the data generated (Appendix A) may serve as a database for studying gene expression changes in skeletal muscle with mixed fiber types in healthy humans without obesity. Our findings suggest that, during disuse, regulation at the transcriptional level plays only a partial role in the down-regulation of sarcomeric and mitochondrial proteins but is closely associated with impairments in lipid transport and beta oxidation (by regulating genes encoding rate limiting enzymes). In line with the typical phenotypic changes, regular aerobic exercise seems to strongly up-regulate genes encoding ECM proteins and many enzymatic ECM regulators, as well as angiogenesis-related genes. By contrast, acute exercise mainly induces the expression of regulatory genes encoding cytokine, growth and secreted factors, and in particular TFs and co-activators. No evidence of a transcriptional mechanism underlying the exercise-induced increase in mitochondrial proteins was found.

Our bioinformatics analysis revealed that several different sets of highly enriched TF binding sites are associated with the transcriptomic changes induced by disuse and regular or acute exercise. It is significant to note that the roles of the majority of these TFs in the context of disuse and exercise have not been studied. The transcriptomic changes induced by disuse and regular exercise are associated with a smaller number of enriched TFs than those induced by acute exercise, suggesting that other mechanisms (e.g., epigenetic modulation) play an important role in regulating gene expression during long-term (chronic) interventions such as disuse and regular exercise training. Transcriptomic responses to acute stress (such as exercise) are strongly associated with the dynamic activation of multiple TFs, most of which are related to the ATF/CREB/AP-1 superfamily and probably act as numerous heterodimers.

Overall, our current study provides a comprehensive analysis of the transcriptomic responses of human skeletal muscle to multidirectional interventions (disuse and aerobic exercise) and reveals a role of transcriptional regulation in the phenotypic changes described in the literature. The transcriptomic responses to chronic interventions (disuse and regular training) partially correspond to the phenotypic effects. Acute exercise (stress) induces changes that are mainly related to the regulation of gene expression, including a strong enrichment of several TFs associated with the up-regulation of gene expression, and a massive increase in the expression levels of genes encoding TFs and co-activators. Overall, the adaptation strategies of skeletal muscle to decreased and increased levels of physical activity differ in direction and demonstrate qualitative differences that are closely associated with the activation of different sets of TFs.

## 4. Materials and Methods

### 4.1. Search Strategy and Eligibility Criteria

Studies investigating the effects of disuse, long-term aerobic exercise training and acute aerobic exercise on transcriptome in the human VL muscle were searched in PubMed and GEO. Eligibility criteria to our meta-analysis were 1) healthy participants without obesity (BMI < 30) and metabolic disorders; 2) existence of raw data; 3) three or more sets of raw data for disuse or training studies or for each recovery time point in acute exercise studies; 4) duration of disuse and training 5 to 21 days and > 5 weeks, respectively (Table 1).

### 4.2. Data Synthesis and Analysis

#### 4.2.1. DNA Microarray Data Processing

Raw data were analyzed using oligo and limma R packages. Background correction was performed using ‘rma’ (Affymetrix, NimbleGen chips) or ‘normexp’ (Agilent, GenePix, Illumina) methods with subsequent quantile normalization. If several probes per a gene were identified then a probe with maximal average expression was taken. Differential expression analysis was performed using the limma R package (analysis of paired samples) with the Benjamini–Hochberg correction. For chips that models are supported by the biomaRt database, Probe IDs were directly annotated by Ensembl gene names (version GRCh38.99) using the biomaRt package. For other chips, manufacturer annotations (available on GEO or in R packages) were used to identify the transcripts IDs, with subsequent annotation by Ensembl gene names by the biomaRt using RefSeq, Entrez, HGNC and European Nucleotide Archive dictionaries.

#### 4.2.2. RNA Sequencing Data Processing

Data processing was described in our previous studies [19,48]. Briefly, adaptor sequences and low-quality reads were trimmed using the Timmomatic tool (v0.36). Reads were aligned to the reference genome GRCh38 (with Ensembl annotation GRCh38.99) and splice junctions were detected using HISAT2 v2.1.0. featureCounts (Rsubread R package) was used to count the numbers of unique reads mapped to known exons of each gene (with Ensembl annotation GRCh38.99). Differential expression analysis was performed using the DESeq2 R package (analysis of paired samples) with the Benjamini–Hochberg correction. To estimate individual fold changes, the normalized read counts (the median normalization) were calculated.

For CAGE data, adapter sequences and low-quality reads were trimmed and 5′-guanine was clipped. Reads were aligned using STAR (v2.7.0a) with mismatch rate threshold 0.06. Uniquely mapped TSS tags (reads) were annotated and counted using bedtools (v2.29) and Ensembl annotation (GRCh38.99).

#### 4.2.3. Differentially Expressed Genes Related with a Specific Intervention (Meta-Analysis)

Genes for each set of raw data were combined into specific matrixes: “Disuse”, “Long-term training” and “Acute exercise 0 h, 0.5–1 h, 3 h, or 4 h”, where the number indicates the time elapsed after the end of exercise (Appendix A). For meta-analysis we integrated gene lists for each intervention in an unbiased manner using the robust rank aggregation method [10]. A gene in a matrix was included into the robust rank aggregation analysis if the gene: (i) showed |Fold Change| > 1.25, P_adj_ < 0.05 for RNA seq and < 0.1 for DNA microarray and (ii) was a translated gene (protein coding genes, polymorphic pseudogene or translated pseudogene). To define differentially expressed genes (DEGs) by the robust rank aggregation method P_adj_ < 0.01 was used (Appendix A). Additionally, reference values of mean gene expression (FPKM) in the vastus lateralis muscle of young healthy males (samples from the GSE120862) were calculated using the Cufflinks (v2.2.1) (Appendix A).

#### 4.2.4. Bioinformatics Analysis

Functional enrichment of biological processes within each set of DEGs related with a specific intervention was performed by the DAVID 6.8 using GOTERM_BP_Direct database (P_adj_ < 0.05 Fisher exact test with Benjamini correction). A set of reference genes was determined for each intervention; only genes evaluated in each transcriptomic dataset were included in the reference gene set. Because enriched Gene Ontology (GO) terms usually demonstrate redundancy, we combined synonymous GO terms into a group using the REVIGO tool (similarity—medium; GO database—Homo sapiens; semantic similarity measure—SimRel; Appendix A).

Complete sets of genes encoding proteins related to glycolysis/gluconeogenesis, lipid transport and beta oxidation, tricarboxylic acid cycle and oxidative phosphorylation were formed using the KEGG; mitochondrial proteins using the Human MitoCarta2.0; collagens, ECM glycoproteins, proteoglycans and ECM regulators using the Human Matrisome; cytokine, growth and secreted factors using the Human Matrisome and the GO BP; transcriptional regulators (transcription factors, co-activators and repressors) using the GO MF and [93] and sarcomeric proteins using the GO BP and the UniProtKB Keywords. Sarcomeric proteins with different concentrations between muscle fiber type I and II were identified using data of a proteomic study [55]. All sets of genes were manually curated; to remove low expressed genes the cutoff FPKM < 0.5 was applied (Appendix A).

The promoter region sequence (from −1500 to +500 of the transcription start site) was used to search for transcription factor binding sites (TFBSs). The transcription start site for each DEG was identified by the coordinates of most abundant transcripts for 28 muscle samples (young males) taken in baseline state (GSE120862 [20]). Taking into account the effect of promoter shift potentially related with acute exercise (e.g., like PPARGC1A gene [94]), the most abundant isoform of each DEG was identified in human (young males) skeletal muscle for 21 muscle samples (young males) taken at 1 h and 4 h after acute aerobic exercise (GSE120862). For each intervention the search for TFBSs and site enrichment analysis were performed separately for up- and down-regulated genes by the GeneXplain platform (function “Search for enriched TFBSs (tracks)”) using a database of position weight matrices (PWM) TRANSFAC v2020.2. Maximal adjusted Fold Enrichment (FE_adj_, statistically corrected odds ratios with a 99% confidence interval) was determined for each matrix (site frequency ≤ 1 per 2000 bp); a set of promoters from 5000 random protein-coding genes was used as a reference set of sequences. FE_adj_ > 1.25 and FDR < 0.05 for the exact Fisher test were set as denoting significantly enriched TFBS (Appendix A). The maximal value of FE_adj_ was used if multiple PWM models were identified for a transcription factor.

Protein–protein interaction networks for different sets of TFs were generated by the String 11.0 with interaction sources Experiments and Databases using the MCL clustering method.

To evaluate the relative number of overlapped motifs in the promoter regions of DEGs, the motif was identified using the GeneXplain platform, with PWM cutoff score for maximal motif rate 1 per 50 kb). Then, the relative number of overlapped motifs (in percent) were calculated (bedtools v 2.29) for each motif overlapping with another motif more than half the length (core + flanking regions).

## Figures and Tables

**Figure 1 ijms-22-01208-f001:**
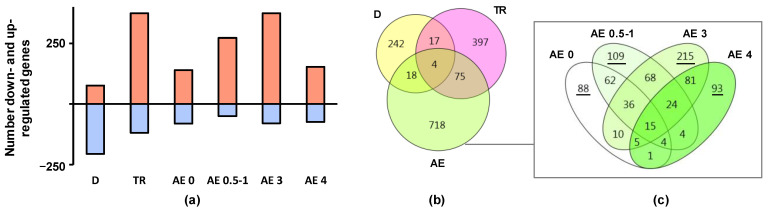
Disuse (D), long-term aerobic exercise training (TR) and acute aerobic exercise (AE) induce multidirectional changes in the expression levels of different sets of genes in the human vastus lateralis muscle. (**a**) The numbers of genes that were up-regulated and down-regulated in each group; (**b**) Overlap between the sets of differentially expressed genes in each group; (**c**) The dynamics of the transcriptional response to acute exercise. The number of unique genes at each time point is underlined.

**Figure 2 ijms-22-01208-f002:**
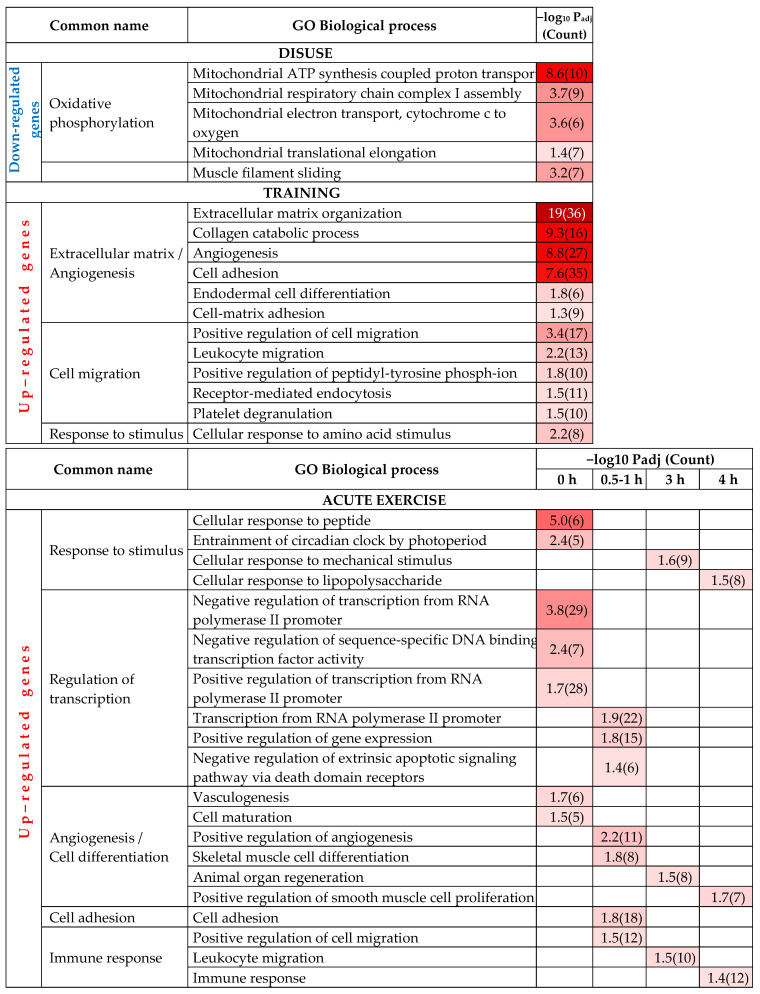
Transcriptomic changes in the human vastus lateralis muscle induced by disuse (D), long-term aerobic exercise training (TR) and acute aerobic exercise (AE) are associated with different biological processes. The heat map shows the P-value adjusted (Padj). Dark red shading denotes the most significant Gene Ontology (GO) terms. The count indicates the number of genes enriched into a term.

**Figure 3 ijms-22-01208-f003:**
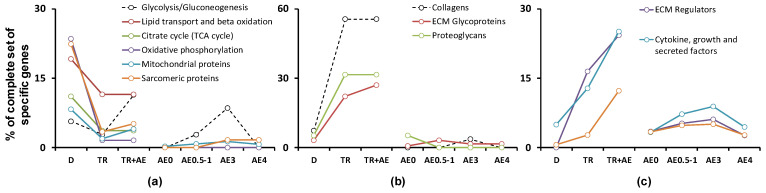
Genes related to different protein classes or metabolic pathways demonstrate different patterns (**a**–**c**) of transcriptional regulation following disuse (D), long-term aerobic exercise training (TR) and acute aerobic exercise (AE). The percentages of the complete sets of specific genes.

**Figure 4 ijms-22-01208-f004:**
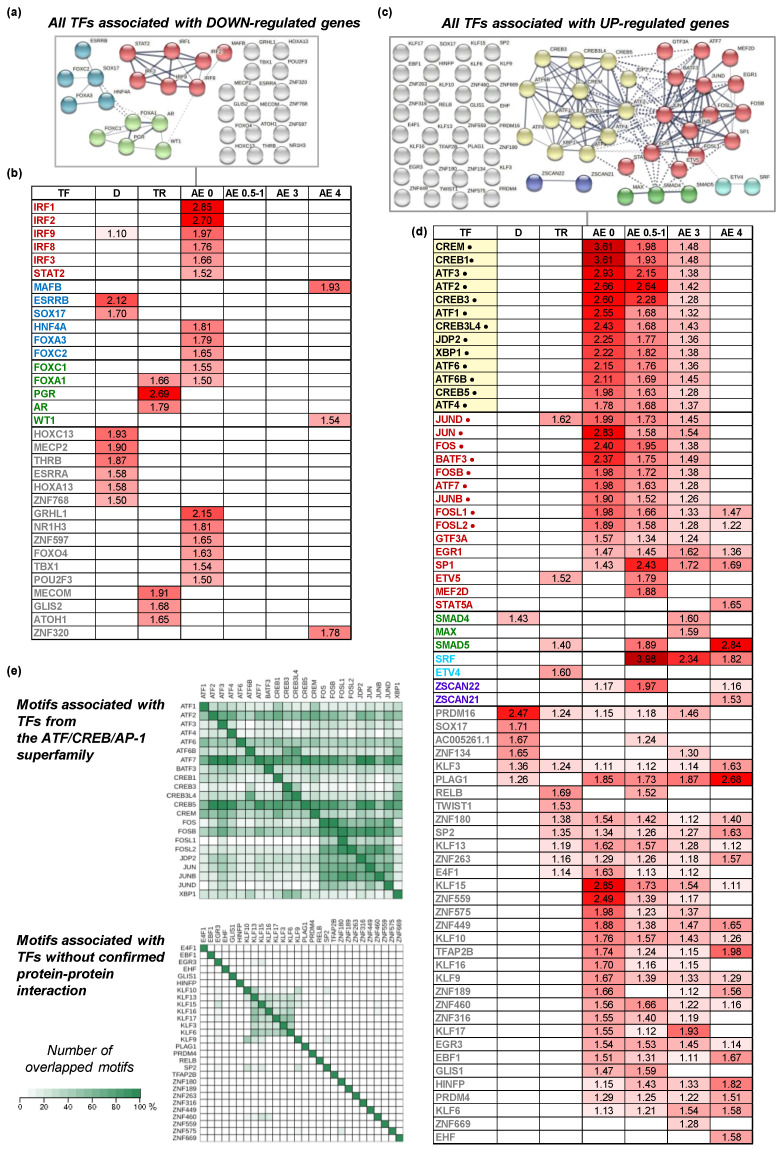
The most enriched (FEadj > 1.5) transcription factors associated with transcriptomic responses to disuse or exercise. (**a**,**c**) A transcription factor binding site enrichment analysis showing transcription factors (TFs) associated with genes that were down-regulated (**a**) or up-regulated (**c**) following each intervention; then a protein–protein interaction analysis was performed to identify clusters (denoted by different colors) of TFs that potentially form dimers or multimers. Darker edges indicate higher interaction scores; (**b**,**d**) The results of a transcription factor binding site enrichment analysis showing TFs associated with genes that were down-regulated (**b**) or up-regulated (**d**) following disuse (D), long-term aerobic exercise training (TR) or acute aerobic exercise (AE) at a specific time of recovery. The heat map and numbers show FEadj (dark red denotes the highest values). TFs related to the ATF/CREB/AP-1 superfamily are marked by •; (**e**) The relative numbers of overlapping motifs associated with bZIP factors from the ATF/CREB/AP-1 superfamily, forming protein–protein interactions (top) and with other factors related to the acute exercise response (bottom).

**Table 2 ijms-22-01208-t002:** Changes in human skeletal muscle phenotype (literature data) and expression of genes encoding proteins of some protein classes (sarcomere, ECM proteins and regulators, cytokine, growth and secreted proteins, transcriptional factors and mitochondrion) or metabolic pathways (fat and carbohydrate metabolism) induced by disuse (D), long-term aerobic exercise training (TR), or both training and acute aerobic exercise (TR+AE) as well as by acute aerobic exercise at specific time of recovery. ↑, ↓, and ↔ show up-, down-regulation, and no changes, respectively.

Metabolic Pathway/Protein Class			D	TR	TR + AE	AE 0 h	AE 0.5–1 h	AE 3 h	AE 4 h
Fiber type specific sarcomeric proteins/genes: **58**		Protein concentration	↔↓ [34]	↔↑ [34,35]				↔↓ [36]	
		*MF1 CSA*	↓ [37,38,39,40,41]	↔ [42]					
		*MF2 CSA*	↓ [37,38,39,40,41]	↔ [42]					
		Genes Up			1			1	
		Genes Down	13	2	2				
		*MF1 specific*	*12*	*1*				*1*	*1*
		*MF2 specific*	*1*	*1*					
Extracellular matrix	Collagen proteins/genes: **27**	Protein concentration	↔ [43,44,45]	↑ [35,46]					
		Protein content	↓	↑					
		Genes Up	2	15	15			1	
		Genes Down							
	ECM Glycoprotein proteins/genes: **122**	Protein concentration							
		Genes Up		26	31	1	4	2	1
		Genes Down	4	1	2				
	Proteoglycan proteins/genes: **19**	Protein concentration		↔↑ [35]					
		Genes Up	1	6	6	1			
		Genes Down							
	ECM Regulator proteins/genes: **115**	Protein concentration		↑ [46,47]					
		Genes Up		19	26	3	6	6	2
		Genes Down			2	1		1	1
Cytokine, growth and secreted factor proteins/genes: **179**		Protein concentration				↑ [1,48]			
		Genes Up	5	20	37	3	11	14	7
		Genes Down	4	3	8	3	2	2	1
Transcriptional regulator proteins/genes: **1527**		Protein concentration							
		Genes Up	6	11	130	42	66	61	26
		Genes Down	4	21	58	10	7	16	16
Glycolysis/ Gluconeogenesis proteins/genes: **35**		Protein concentration	↔↓ [34,41,49]	↔↑ [35,36,50]				↔↓ [36]	
		Genes Up		1	4		1	3	
		Genes Down	2						
Lipid transport and beta oxidation proteins/genes: **26**		Protein concentration	↓ [41,49]	↑ [35,51]					
		Genes Up		3	3				
		Genes Down	5						
Mitochondrion	Citrate cycle (TCA cycle) proteins/genes: **27**	Protein concentration	↓ [34,41,49]	↑ [35,36,50]				↔↓ [36]	
		Genes Up		1	1				
		Genes Down	3						
	Oxidative phosphorylation proteins/genes: **123**	Protein concentration	↓ [34,41,49]	↑ [35,36,50]				↔↓ [36]	
		Genes Up		2	2			0	
		Genes Down	29						
	Mitochondrial proteins/genes: **1097**	Protein concentration	↓ [34,41,49]	↑ [35,36,50]				↔↓ [36]	
		Genes Up	3	18	35	2	6	11	8
		Genes Down	88	3	9	1	3	4	

## Data Availability

The data presented in this study are available in Appendix A here.

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
