# Peer review of "Transcriptomic Signatures and Upstream Regulation in Human Skeletal Muscle Adapted to Disuse and Aerobic Exercise"

_ijms, 2021, doi:10.3390/ijms22031208_

Round 1

Reviewer 1 Report

It is a well-designed meta-analysis and the authors have included recent studies and included 27-transcriptomic datasets. The screening of data and presentation sounds very good and the aim of the study to analyze the effects of inactivity and aerobic exercise on human skeletal muscle reflects the changes in gene regulation. The authors have presented transcriptomic changes in the human vastus lateralis muscle induced by inactivity long-term aerobic exercise training, and acute aerobic exercise associated with different biological processes such as upregulated genes with different biological functions such as Oxidative phosphorylation, Regulation of transcription, Immune response and many more functions. The manuscript is well written. I appreciate the design of the study and the manuscript could be accepted with minor modification.

  1. Rephrase the sentence in line #30 ‘Skeletal muscles constitute a third of the human body’.
  2. Check the formatting error of table 1.
  3. Figure 4a and 4c are blurred, replaced with high resolution one.

Reviewer 2 Report

Makhnovskii and colleagues have undertaken a meta-analysis of the transcriptomic signature related to inactivity and acute and regular aerobic exercise to identify differentially expressed genes and transcription factors associated with each intervention. The authors note the limitations and low concordance of previous studies, and their robust meta-analytical approach adds novel and impressive data to the field in this regard. The paper was very well written and interesting to read. I have minor comments for consideration by the authors below:

  1. “Skeletal muscle” and “exercise” should be added to the keywords section.
  2. There is a font size issue in the introduction from lines 58-78 (a paragraph has been incorporated into the Table 1 legend. There are also formatting issues in lines 80-81.
  3. Table 1: Concerning the studies that were included in the meta-analysis as outline in Table 1, could the authors please further define what is meant by limb immobilization versus limb suspension as a model? And compared to the bed rest model, to what extent are the subjects in the limb immobilisation and suspension models still able to activate the rest of the body? For example, myokines/cytokines and immune cells from distant active muscle or non-muscle tissues may still be able to signal to the musculature of the immobilised limb if whole body activity is significant. The inclusion of these different models that represent "inactivity" need to be justified.
  4. Table 1: To what extent do the transcriptome data differ between the models of exercise included in the study i.e. cycling versus knee extension versus running? As per Comment 3 above, running would incorporate most of the body’s muscle mass but without the resistance of cycling or weight loading in knee extension exercises, for example. Again, the inclusion of these models as "exercise" need to be justified.
  5. Table 1: Were there noticeable differences in the transcriptomes for males versus females for all groups? This should probably be mentioned in the methods, where is appears for some data sets, females were excluded.
  6. I commend the authors on the inclusion of a Limitations section. Here, they discuss many of the thoughts I had after reading the paper e.g. the molecular response of muscle during re-adaptation after a period of inactivity. To this effect, I am wondering where sedentary individuals lie in the scope of the findings presented in this manuscript too i.e. people who do not partake in significant exercise but who are mobile and weight bear? How far away from “inactive” responses are the skeletal muscles of these individuals likely to be, especially in view of the similar gene expression profiles between the inactive group and sarcopenic muscle mentioned in the discussion?
  7. Onward from Comment 6 - could the authors comment on whether there could be differences in the transcriptomes of deep versus the superficial (i.e. vastus lateralis) skeletal muscles in response to exercise…particularly leg centric exercise such as cycling and knee extension? There are some papers that suggest deep versus superficial muscles might differ in their response to an exercise bout.
  8. Methods: It is mentioned in in lines 494 and 495 that DEG isoforms were identified exclusively in males. Please justify why females were excluded from these analyses since females were shown to be included in the study in Table 1.
  9. Minor grammatical issues:
    • The sentence beginning on line 72 and ending on line 73 needs to be revised for clarity/grammar.
    • Replace “as” with “at” in line 198.
    • Line 276 needs to be revised for clarity.
    • Insert “the” in between “in” and “baseline” in line 361.
